

# Seasonal variation in microhabitat of salamanders: environmental variation or shift of habitat selection?

Enrico Lunghi[1,2,3], Raoul Manenti[4] and Gentile Francesco Ficetola[5,6,7]

[1] Universität Trier Fachbereich VI Raum- und Umweltwissenschaften Biogeographie, Campus I, Gebäude N Universitätsring, Trier, Germany
[2] Natural Oasis, Prato, Italy
[3] Museo di Storia Naturale dell'Università di Firenze, Sezione di Zoologia "La Specola," Firenze, Italy
[4] Dipartimento di Bioscienze, Università degli Studi di Milano, Milano, Italy
[5] Dipartimento di Scienze dell'Ambiente e del Territorio e di Scienze della Terra, Università degli Studi di Milano-Bicocca, Piazza della Scienza, Milano, Italy
[6] Laboratoire d'Ecologie Alpine (LECA), Université Grenoble-Alpes, Grenoble, France
[7] CNRS, LECA, Grenoble, France

Corresponding author
Enrico Lunghi,
enrico.arti@gmail.com

## ABSTRACT

Relationships between species and their habitats are not always constant. Different processes may determine changes in species-habitat association: individuals may prefer different habitat typologies in different periods, or they may be forced to occupy a different habitat in order to follow the changing environment. The aim of our study was to assess whether cave salamanders change their habitat association pattern through the year, and to test whether such changes are determined by environmental changes or by changes in preferences. We monitored multiple caves in Central Italy through one year, and monthly measured biotic and abiotic features of microhabitat and recorded Italian cave salamanders distribution. We used mixed models and niche similarity tests to assess whether species-habitat relationships remain constant through the year. Microhabitat showed strong seasonal variation, with the highest variability in the superficial sectors. Salamanders were associated to relatively cold and humid sectors in summer, but not during winter. Such apparent shift in habitat preferences mostly occurred because the environmental gradient changed through the year, while individuals generally selected similar conditions. Nevertheless, juveniles were more tolerant to dry sectors during late winter, when food demand was highest. This suggests that tolerance for suboptimal abiotic conditions may change through time, depending on the required resources. Differences in habitat use are jointly determined by environmental variation through time, and by changes in the preferred habitat. The trade-offs between tolerance and resources requirement are major determinant of such variation.

Subjects Animal Behavior, Ecology, Zoology
Keywords Tolerance, *Hydromantes italicus*, Cave, Distribution, Microclimate, Physiological niche, Spatial segregation, Biospeleology, Plethodontid

## INTRODUCTION

The use of habitat models to evaluate factors determining species distributions is becoming increasingly prevalent in ecological research (*Peterson et al., 2011*; *Warren, 2012*; *Stein, Gerstner & Kreft, 2014*). Such models help understanding the factors determining species occurrence, and may allow predicting potential areas of occupancy, with important consequences for planning adequate conservation actions (*Domíguez-Vega et al., 2012*; *Bogaerts et al., 2013*). Despite repeated calls for mechanistic modelling (*Kearney & Porter, 2009*), correlative habitat models remain the most frequently used approach. Correlative models combine data on species occurrence (e.g., presence/absence, presence-only, abundance) with information on environmental features, identifying statistical relationships which represent the basis for model predictions (*Guisan & Thuiller, 2005*). Such models are based on the assumptions that species presence is associated with favorable environmental features (species-habitat association) (*Godsoe, 2010*).

However, patterns of species-habitat association may be not consistent during time. Analyses of habitat associations generally assume that species are at quasi-equilibrium with the environment, but this assumption may not always hold (e.g., during dispersion or contraction phases) (*Saupe et al., 2014*). Furthermore, differences in habitat association patterns may occur through two distinct, non-exclusive processes: the species may select different habitats across their life-time (selection change hypothesis), and environmental features may change through time (environmental change hypothesis). According to the selection change hypothesis, a given species may be associated with different environmental features in different time periods and/or life stages. For instance, many species show seasonal activities, and select different environments depending on the activities performed (e.g., nesting, foraging, wintering) (*Seebacher & Alford, 1999*; *Brambilla & Saporetti, 2014*). In the long term, temporal variation for habitat association in a given species may also occur due to evolution of novel adaptations (*Nogués-Bravo, 2009*; *Stigall, 2012*). According to the environmental change hypothesis, temporal variation that exists for the many biotic and abiotic features can affect species distribution (*Kearney et al., 2013*). Such variation may occur over both short (e.g., variation of vegetation cover or temperature among the seasons) and longer timescales (e.g., climate change, habitat degradation) (*Saupe et al., 2014*). Both selection changes and environmental changes may influence the possibility of predicting species distribution in different time periods. Evaluating whether habitat association pattern changes through time, and the factors determining such variation, is extremely important to assess the transferability and generality of conclusions drawn from habitat modeling.

Among amphibians, plethodontid salamanders represent a very interesting study case. Due to their particular physiology, they need a narrow combination of environmental characteristics, and actively search places with suitable microclimatic conditions (cold temperature and high moisture; *Spotila, 1972*; *Camp & Jensen, 2007*. Cave salamanders (genus *Hydromantes*) may live both in surface and subterranean environments, but must move underground during the arid and hot Mediterranean summer, when the surface conditions become hot and dry (*Lanza et al., 2006*; *Ficetola, Pennati & Manenti, 2012*).

 

In subterranean environments microclimatic features are often considered to remain approximately stable, giving organisms the opportunity to inhabit caves constantly. Some studies have shown that cave salamanders are associated with caves having specific environmental features, such as low temperature, high humidity and presence of prey (*Ficetola, Pennati & Manenti, 2012*; *Lunghi, Manenti & Ficetola, 2014*), but these studies have been often performed during summer, when outdoor conditions are particularly unsuitable for salamanders, and abundance in cave is highest. However, caves are not closed systems, and environmental characteristics within caves can change over time due to external influences (*Romero, 2009*). Such fluctuations mostly affect areas near the entrance of caves (twilight zone) and can strongly influence cave communities (*Ficetola, Pennati & Manenti, 2013*; *Camp et al., 2014*; *Lunghi, Manenti & Ficetola, 2014*). Nevertheless, the few studies analyzing the seasonal variation in the distribution of European cave salamanders (*Salvidio et al., 1994*; *Vignoli, Caldera & Bologna, 2008*) did not test whether habitat selection changes through time.

The peculiar features of both caves and plethodontid salamanders make them an excellent system for species-habitat association studies. Cave environments are dominated by few, simple environmental gradients, such as light, depth, temperature, humidity and food availability (*Romero, 2009*), affording simplistic habitat characterization. Furthermore, species are easily detectable inside the delimited cave environments (*Ficetola, Pennati & Manenti, 2012*), allowing a reliable identification of occupied and unoccupied sectors.

The aim of this study was analyzing the variation through time of species-habitat association in the Italian cave salamander (*Hydromantes italicus*). First, we used habitat models to identify the relationships between the distribution of salamanders and microhabitat features, evaluating if the pattern of microhabitat association is constant through time. Second, we assessed whether the temporal variation in microhabitat occurs because the species selects different environmental features through the year, or because habitat features are affected by seasonal variation (i.e., we evaluated the support of the environmental changes vs. selection change hypotheses).

## MATERIAL AND METHODS

### Authorizations

All applicable institutional and/or national guidelines for the care and use of animals were followed. The study was conducted under authorization of Apuan Alps Regional Park (no 5, 4/04/2013), District of Prato (no 448, 2013), District of Pistoia (no 0022597/2013/P) and District of Lucca (no 731, 21/02/2013).

### Surveys

For 12 months (from January 2013 to December 2013) we monitored 15 caves occupied by *Hydromantes italicus* in the North of Tuscan Apennines (Central Italy, between 43°52′42″N, 11°07′18″E and 43°59′51″N, 10°13′48″E). Preliminary surveys performed in 2012 indicated the presence of *H. italicus* at all sites. Surveys were conducted during day-time. The order of cave survey was chosen randomly, and the time interval between

successive visits was 9–45 days. During surveys, for each cave we recorded monthly environmental data both inside and outside caves. Outside caves, we registered air temperature (accuracy: 0.1 °C) and humidity (accuracy: 0.1%) using a thermo-hygrometer Lafayette TDP92, in a shaded area 5–10 m from the entrance. The interior of each cave was divided into sectors of 3-m length, starting from the entrance and extending to the deepest explored area: our exploration was conducted until the end of the caves, or until the deepest sector reachable without speleological equipment. Three-m sectors approximately correspond to the home range of *Hydromantes* during their hypogean activity (*Salvidio et al., 1994*). Overall, we recorded data from 121 cave sectors [average development explored per cave: 24.2 m (range 6–60), corresponding to 2–20 sectors per cave]. At the time of surveys, in each sector we recorded four parameters known to influence cave salamanders. Air temperature, humidity and incident light (illuminance, measured using a Velleman DVM1300 light meter, minimum recordable light: 0.1 lux) represented the abiotic conditions of caves, which influence metabolism, water balance and activity (*Kearney et al., 2013*). The abundance of *Meta menardi* spiders was considered as a biotic variable. On the one hand, *Meta* spiders are major predators of juvenile salamanders (*Lanza et al., 2006*). Furthermore, *Meta* spiders are associated with areas showing high invertebrate abundance, and have been proposed as an indicator of prey abundance in cave environments (*Manenti, Lunghi & Ficetola, in press*). See *Ficetola, Pennati & Manenti (2013)* for additional details on the recording of cave features.

We used visual encounter surveys to assess the presence/absence of *H. italicus* and *M. menardi* spiders in each sector. This standardized technique allows to verify the presence of species in an area during a defined time (*Crump & Scott Jr, 1994*; *Jung et al., 2000*). If possible, salamanders were measured. Salamanders showing total length >6.5cm or with male secondary characters were considered adults (*Lanza et al., 2006*), the remaining salamanders were considered juveniles. All individuals were immediately released at the collection point.

## Statistical analyses
### Variation of environmental features of caves
We used linear mixed models (LMM) to analyze the temporal variation of cave microhabitat. We used the Akaike's Information Criterion corrected for small sample size (AICc) to identify the combination of parameters that better explain the variation of microclimatic features inside caves (*Stephens et al., 2007*). In LMM, we considered cave features (temperature, humidity, illuminance) as dependent factors, while outdoor features (temperature and humidity), linear distance from the cave entrance (hereafter, depth) and month of survey were considered as independent factors. We also considered the interaction between depth and month of survey. We also included the time of survey (hour and minute in which we began the survey) as an additional independent variable. Cave and sector identity were considered as random categorical variables, as they shows a typical combination of variables (both biotic and abiotic) independently from their position and location. For all models, Variance Inflation Factor was <5, confirming lack
of collinearity issues (*Fox, 2002*). Seasonal variation also occurs for the distribution of cave spiders but was not analyzed here as it will be the focus of a separate study.

### Relationships between species and environmental features

Not detecting a species during a survey does not necessarily mean that species is absent, as most species have detection probability <1 (*MacKenzie et al., 2006*). Standard approaches to the analysis of detection probability assume that sites are closed to changes in the state of occupancy for the duration of sampling (*MacKenzie et al., 2006*). However, cave salamanders quickly modify their occupancy patterns throughout the year in response to environmental variation (*Briggler & Prather, 2006*; *Camp & Jensen, 2007*; *Vignoli, Caldera & Bologna, 2008*), and therefore violate the closed population assumption. Approaches assuming open populations also exist but, in this study case, their implementation would require assumptions on population dynamics for which no data were available (*Dail & Madsen, 2011*). Sampling effort was standardized across sectors. Therefore, following recommendations by *Banks-Leite et al. (2014)*, we preferred performing analyses using standard mixed models, while verifying that low detection probability did not bias our results.

First, we used generalized linear mixed models (GLMM) assuming binomial error to identify the relationships between the presence of salamanders and environmental features (air temperature, humidity, illuminance and spider abundance) of each sector, throughout the 12 months of sampling. To assess whether the habitat selection pattern is constant through time, we included the interactions between sampling month and environmental features. Sector and cave identity were included as random categorical factors. We built all possible model combinations, and ranked them using AICc. Complex models with AICc values higher than the simpler, nested models were not considered as candidate models (*Richards, Whittingham & Stephens, 2011*). We used a likelihood ratio test to assess the significance of terms in the best-AICc model. As microhabitat selection may be different among age classes (*Ficetola, Pennati & Manenti, 2013*), this analysis was repeated three times: first, considering all individuals, then considering adults and juveniles separately.

The results of the previous models may be affected by imperfect detection. We used the *MacKenzie & Kendall (2002)* approach to test detection probability of cave salamanders, on the basis of data collected in 22 sectors from three different caves. These caves were surveyed in late June-early July: during this interval, *Hydromantes* movements among sectors are expected to be limited (*Lanza et al., 2006*). For these sectors, two surveys were performed 9–14 days apart, therefore we assumed constant occupancy in this interval and estimated detection probability using single-season closed population occupancy models with the unmarked package in R (*Fiske & Chandler, 2011*). The analysis of detection probability was repeated twice: assuming constant detection across sectors, and assuming that detection probability is related to distance from cave entrance. We then used AIC to identify the best detection probability model.

Analyses (see results) showed a per-visit detection probability of 0.75, i.e., two surveys allow to ascertain presence/absences with 94% confidence (*Sewell, Beebee & Griffiths, 2010*). To assess the robustness of habitat models to imperfect detection, we also repeated the GLMM analysis by comparing two contrasting periods seasons: January–February

and June–July. Movements between superficial and deep sectors are more frequent during spring and autumn (*Lanza et al., 2006*), thus we assumed that occupancy was relatively stable within these periods. We merged data from two-months periods respectively into winter (January–February) and summer (June–July), and repeated the analyses using the same variables of the best-AICc models obtained from the analyses of full dataset.

### Testing the stability of habitat selection pattern

We used niche equivalency tests to assess whether salamanders select sectors with similar environmental features in different months, after taking into account differences for the availability of microhabitat conditions (*Broennimann et al., 2012*). The similarity of the habitat selection pattern in two distinct seasons was assessed using Schoener's *D*, a metric of niche similarity (*Warren, Glor & Turelli, 2008*; *Saupe et al., 2014*). For equivalency tests, salamander occurrences from different months were pooled and then randomly split in two datasets, maintaining the same number of occurrences of the original datasets; Schoener's *D* was then calculated. This procedure was repeated 300 times to assess whether niche similarity was significantly lower than expected by chance. The equivalency test was repeated for the two environmental variables (temperature and humidity) for which habitat models suggested differences among months. We focused on univariate rather than multivariate tests because we were interested on variation of habitat selection due to change of specific variables (*Saupe et al., 2014*). This analysis was performed on four months (January, February, June and July) showing contrasting patterns of habitat association (see results), and during which we do not expect major movements among cave sectors (i.e., within these intervals the quasi-equilibrium assumption is more likely to be hold than when seasonal migrations occur). The analysis was performed on all individuals together and for each age category (juveniles only and adults only). Since six pairwise tests were performed for each group and for each variable, significance values were corrected using sequential Bonferroni's correction (*Rice, 1989*).

## RESULTS

### Variation of environmental features inside caves

Internal temperature was strongly related to external temperature and humidity, month, depth and interaction between month and depth: all variables except depth were significant (Tables 1A and 2A). Seasonal change led to thermal inversion inside caves: from late autumn to early spring temperature increased with depth, while from late spring to early autumn temperature decreased in the deep sectors (Fig. 1A). Humidity inside caves was strongly related to external humidity, month, depth and to the time of survey. Furthermore, the significant interaction between month and depth indicated that the humidity gradient was not constant through the year (Tables 1B and 2B). The deepest sectors showed high stability of humidity through time, while fluctuations due to external variation were evident in sectors nearby the cave entrance. External humidity was particularly high in autumn and spring, determining an increase of humidity in the first sector of caves (Fig. 1B). Internal light incidence was related to depth and external humidity (Tables 1C and 2C). The deepest sectors always showed lower light

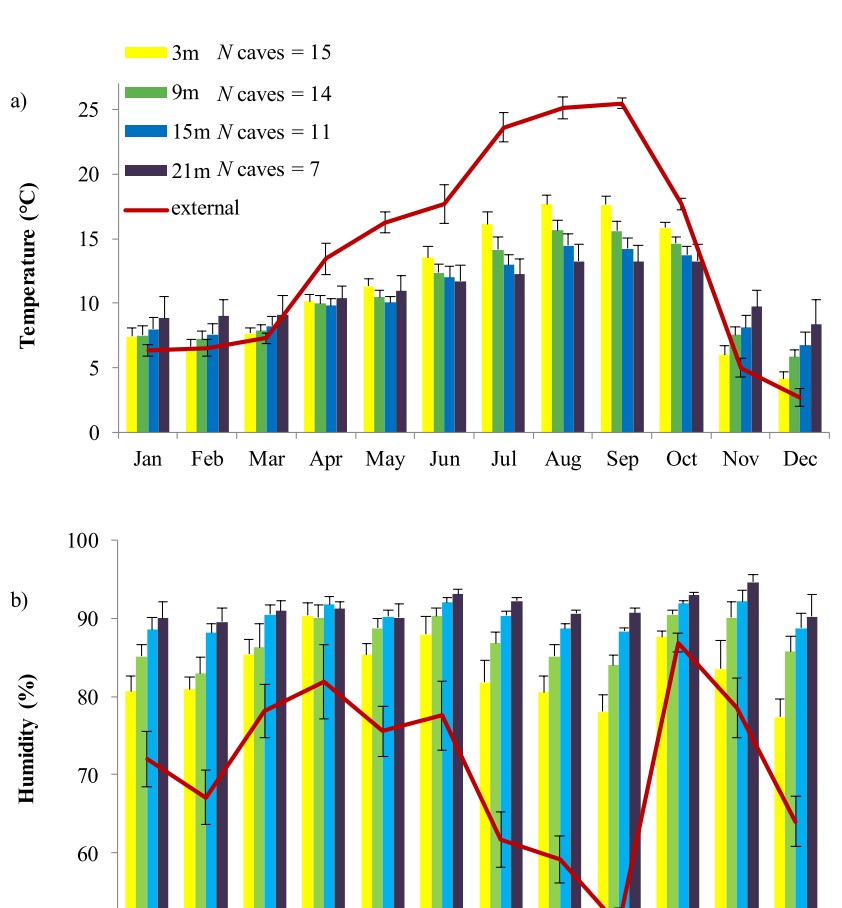

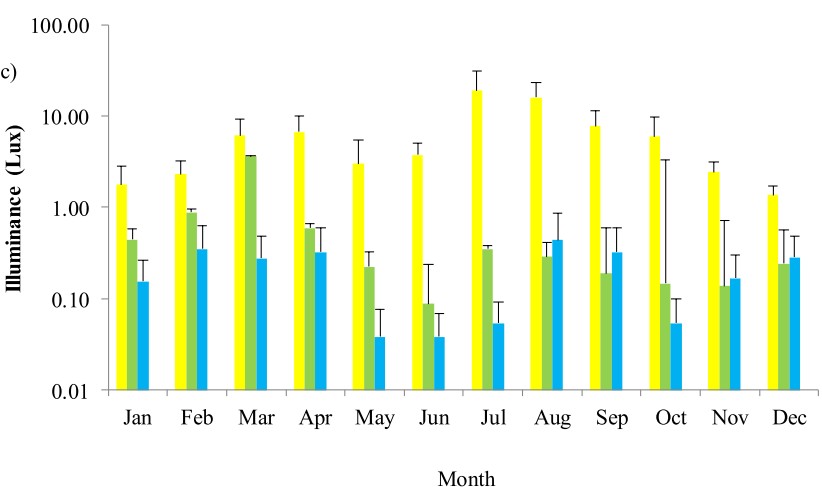

**Figure 1 Annual variation of external environment and cave microhabitat.** Internal variables are (A) temperature, (B) humidity and (C) illuminance (lux). In each graph, colored plots represent sectors located at different distance from the entrance (from 3 to 21 m). These sectors represent the area in which microclimate variability is higher; at 21 m illuminance was constantly 0 lux. Error bars are standard errors. For temperature and humidity, the trend of the respective external feature is also shown, represented by a continuous red line.

**Table 1 Best AIC models explaining the variation in microhabitat features of caves.** We considered as dependent variables inner abiotic features of caves: (A) Temperature, (B) Humidity and (C) Illuminance. We used as independent variables: Month of survey, Time in which the survey began, Depth of sector, External Temperature, External Humidity and interaction between Month and Depth (Prof : M). For each continuous variable, the regression coefficient is reported if the variable is included into a given model. For both categorical variables and interactions, + indicates their presence into the model. For each independent variable, we report the first five best models.

| Independent variables included into the model | | | | | | df | AICc | Δ − AICc | Weight |
|---|---|---|---|---|---|---|---|---|---|
| Month | Time of survey | Depth | External temperature | External humidity | Prof : M | | | | |
| (A) Temperature | | | | | | | | | |
| + | | 0.07 | 0.22 | 0.02 | + | 29 | 5,703 | 0 | 0.825 |
| + | 0.05 | 0.07 | 0.21 | 0.02 | + | 30 | 5,706.3 | 3.34 | 0.155 |
| + | | 0.07 | 0.18 | | + | 28 | 5,710.5 | 7.50 | 0.019 |
| + | 0.01 | 0.07 | 0.18 | | + | 29 | 5,717.6 | 14.67 | 0.001 |
| + | 0.12 | 0.07 | | | + | 28 | 5,811.7 | 108.78 | 0 |
| (B) Humidity | | | | | | | | | |
| + | 0.34 | 0.27 | | 0.15 | + | 29 | 8,314.1 | 0 | 0.517 |
| + | 0.31 | 0.27 | 0.11 | 0.16 | + | 30 | 8,314.2 | 0.15 | 0.480 |
| + | 0.26 | 0.24 | 0.15 | 0.17 | | 19 | 8,325.2 | 11.09 | 0.002 |
| + | | 0.27 | 0.16 | 0.15 | + | 29 | 8,329.8 | 15.75 | 0 |
| + | 0.31 | 0.24 | | 0.16 | | 18 | 8,329.8 | 15.75 | 0 |
| (C) Illuminance | | | | | | | | | |
| | | −0.04 | | −0.01 | | 6 | 3,232.3 | 0 | 0.967 |
| | | −0.04 | 0.01 | −0.01 | | 7 | 3,239.7 | 7.41 | 0.024 |
| | −0.01 | −0.04 | | −0.01 | | | 3,241.7 | 9.40 | 0.009 |
| | −0.01 | −0.04 | 0.01 | −0.01 | | | 3,248.7 | 16.39 | 0 |
| | | −0.04 | 0.01 | | | | 3,256 | 23.72 | 0 |

than the superficial ones. However, incident light increased in summer and during periods characterized by low humidity (Fig. 1C).

## Detection of cave salamanders

Through the 180 cave surveys, we obtained 1,087 detections of cave salamanders (289 adult males, 393 adult females, 49 not sexed adults and 356 juveniles). The average sampling effort was of 7.5 min/sector. Salamanders were detected throughout the year with 13% of detections in winter, 39% in spring, 30% in summer and 18% in autumn months. The model assuming constant detection probability across sectors showed a lower AIC value (AIC: 53.3) than the model assuming that detection probability is related to distance from the entrance (AIC: 53.9). Detection probability of salamanders within sectors was high (detection probability ± SE: 0.75 ± 0.12).

## Analysis of occurrence of *H. italicus* through the year

Presence of *H. italicus* was strongly related to month, and was generally associated with sectors characterized by high humidity, low light and abundant *M. menardi* spiders

**Table 2  Parameters related to microclimatic change of caves through the year: best-AICc models.** The dependent variables were three major features of cave microclimate: (A) internal temperature, (B) internal humidity and (C) illuminance. Independent variables were: Month of survey, Depth of sector, Temp. ext (external temperature), Hum. ext (external humidity), Time (hour of survey).

| Factor | $B$ | $\chi_1^2$ | $P$ |
|---|---|---|---|
| (A) Temperature (internal) | | | |
| Month | | 151 | <0.001 |
| Depth | 0.07 | 0.81 | 0.368 |
| Temp. ext | 0.21 | 144.2 | <0.001 |
| Hum. ext | 0.02 | 18.96 | <0.001 |
| Month × depth | | 680.71 | <0.001 |
| (B) Humidity (internal) | | | |
| Month | | 117.03 | <0.001 |
| Depth | 0.27 | 105.91 | <0.001 |
| Hum. ext | 0.16 | 205.3 | <0.001 |
| Time | | 27.95 | <0.001 |
| Month × depth | | 94.92 | <0.001 |
| (C) Illuminance | | | |
| Depth | −0.03 | 34.60 | <0.001 |
| Hum. ext | −0.01 | 49.66 | <0.001 |

(Tables 3A and 4A). Furthermore, significant interactions between month and temperature and between month and humidity indicated different microhabitat selection patterns among months (Table 4A). Specifically, in winter periods salamanders were associated with warmest sectors, while in summer periods they were associated with coldest and most humid sectors (Figs. 2A and 2B).

The microhabitat selection pattern was similar if adults only were considered. Adults were more abundant in sectors with low light and abundant *M. menardi* (Tables 3B and 4B). Furthermore, differences among months were strong, and the interactions between month and both humidity and temperature were significant. Adults were associated with relatively cold sectors during summer, while in winter they were associated with warmer sectors (Fig. 2C). In summer, adults were associated with the most humid sectors; however, they showed a clear preference for the most humid sectors also in February (Fig. 2D).

Juveniles were more frequent in sectors with high humidity and abundant *M. menardi* spiders; furthermore the effect of month, and the interactions humidity-month and temperature-month were significant (Tables 3C and 4C). Juveniles were associated with the coldest sectors during winter and with warmer sectors during spring (Fig. 2E). From late winter until spring, juveniles were associated with sectors characterized by lower humidity, while during summer this apparent preference shifted in favor of most humid sectors (Fig. 2F).

Table 3 **Five best AIC models relating salamander distribution to environmental features.** We considered as dependent variable the presence of (A) the species, (B) presence of Adults and (C) presence of Juveniles. We used as independent variables: internal humidity (Humid), Month of survey, illuminance (Lux), *Meta* spiders abundance and internal temperature (Temp). Furthermore, we also used as independent variables interaction between month and internal humidity (Hum : M), month and illuminance (Lux : M), month and *Meta* spiders (*Meta* : M) and month and internal temperature (Temp : M). For each continuous variable, the regression coefficient is reported if the variable is included into a given model. For categorical variables and interactions, + indicates that the variable or the interaction is included into the model.

| | | | | Independent variables included into the model | | | | | df | AICc | $\Delta - $AICc | Weight |
|---|---|---|---|---|---|---|---|---|---|---|---|---|
| **Humid** | **Month** | **Lux** | *Meta* | **Temp** | **Hum : M** | **Lux : M** | *Meta* : M | **Temp : M** | | | | |
| (A) Presence of the species | | | | | | | | | | | | |
| **1.12** | **+** | **−0.34** | **0.44** | **0.27** | **+** | | | **+** | **40** | **1,384.8** | **0** | **0.709** |
| 1.64 | + | −0.36 | | 0.26 | + | + | | + | 39 | 1,388.9 | 4.15 | 0.089 |
| −2.47 | + | −20.74 | 0.45 | 0.25 | + | | | + | 51 | 1,389.3 | 4.5 | 0.075 |
| 1.41 | + | | 0.45 | 0.27 | + | | | + | 39 | 1,390.2 | 5.45 | 0.046 |
| 7.79 | + | −0.35 | 0.43 | | + | | | | 28 | 1,392.1 | 7.35 | 0.018 |
| (B) Presence of adults | | | | | | | | | | | | |
| **1.27** | **+** | **−0.43** | **0.39** | **0.16** | **+** | | | **+** | **40** | **1,253.8** | **0** | **0.721** |
| 1.7 | + | −0.44 | | 0.16 | + | | | + | 39 | 1,256.2 | 2.44 | 0.213 |
| 1.67 | + | | 0.42 | 0.16 | + | | | + | 39 | 1,261 | 7.25 | 0.019 |
| 6.83 | + | −0.42 | 0.4 | | + | | | | 28 | 1,261.5 | 7.78 | 0.015 |
| 1.46 | + | −0.44 | −0.15 | 0.18 | + | | + | + | 51 | 1,262.7 | 8.92 | 0.008 |
| (C) Presence of juveniles | | | | | | | | | | | | |
| **1.46** | **+** | | **0.61** | **0.41** | **+** | | | **+** | **39** | **807.2** | **0** | **0.428** |
| 1.23 | + | −0.26 | 0.58 | 0.4 | + | | | + | 40 | 807.3 | 0.1 | 0.407 |
| 2.14 | + | −0.3 | | 0.39 | + | | | + | 39 | 810.5 | 3.22 | 0.085 |
| 2.57 | + | | | 0.39 | + | | | + | 38 | 810.8 | 3.57 | 0.072 |
| −2.85 | + | −20.8 | 0.59 | 0.35 | + | + | | + | 51 | 816 | 8.72 | 0.005 |

## Analysis between contrasting seasons

As detection probability was imperfect, we repeated the analysis by focusing on the comparison between two contrasting seasons (winter/summer), in which migration of salamanders is probably limited. During these two intervals we observed 112 salamanders in winter and 257 salamanders in summer. The results of this analysis were generally consistent with the analysis of the full dataset. If all individuals were pooled, salamanders were associated with the darkest sectors. Season strongly affected the presence of salamanders; furthermore, we detected a significant interaction between temperature and season; the interaction between humidity and season was marginally not significant (Table S1A). During these two seasons, salamanders were generally associated with sectors in which microclimate was different from outdoor climate conditions: in fact, they were associated with the most humid and cold sectors during summer, while in winter they preferred relatively warm sectors (Figs. S1A and S1B). Results were nearly identical in the analysis of adults-only (Table S1B; Figs. S1C and S1D). In the analysis of juveniles, only the interaction between season and temperature remained significant (Table S1C, Figs. S1E and S1F). However, it should be remarked that sample size was relatively small in this latter analysis (112 juveniles observed), and this may have limited statistical power, compared to the previous analyses.
**Table 4 Parameters related to presence/absence of salamanders.** The dependent variables were the presence of (A) Species, (B) Adults only and (C) Juveniles only. See Table 1 for explanation of variable names. Only the best-AICc models are shown.

| Factor | B | $\chi^2_1$ | P |
|---|---|---|---|
| **(A) Species** | | | |
| Month | | 140.2 | **<0.001** |
| Humidity | −2.65 | 4.3 | **0.039** |
| Lux | −20.79 | 7.6 | **0.006** |
| *Meta* abund. | 0.36 | 6.3 | **0.012** |
| Temperature | 0.25 | 1.4 | 0.238 |
| Hum × month | | 30.6 | **0.001** |
| Temp × month | | 31.2 | **0.001** |
| **(B) Adults** | | | |
| Month | | 128.7 | **<0.001** |
| Humidity | −1.57 | 1.4 | 0.233 |
| Lux | −1.95 | 9.4 | **0.002** |
| *Meta* abund. | −2.31 | 4.6 | **0.033** |
| Temp | 1.74 | 0.3 | 0.567 |
| Hum × month | | 37.3 | **<0.001** |
| Temp × month | | 32.7 | **<0.001** |
| **(C) Juveniles** | | | |
| Month | | 37.8 | **<0.001** |
| Humidity | −3.60 | 5.4 | **0.02** |
| *Meta* abund. | 0.75 | 5.7 | **0.017** |
| Temp | 0.35 | 3.6 | 0.059 |
| Hum × month | | 37 | **<0.001** |
| Temp × month | | 39 | **<0.001** |

## Stability of habitat selection pattern

Most of equivalency tests were not significant, suggesting that habitat selection pattern was consistent through months (Table 5). However, in the analyses of humidity considering all individuals and juveniles only, niche equivalency was significantly lower than expected by chance between February and June, and between February and July. Salamanders were more tolerant for low-humidity habitats than during winter (Fig. 3). Conversely, if adults only were analyzed, none of similarity tests were rejected (Table 5).

## DISCUSSION

Caves are often described as stable environments (*Romero, 2012*), but their features and the distribution of their inhabitants shows strong fluctuations through the year, particularly in the superficial sectors. No doubt, the strong seasonal variation of salamander distribution was mostly dictated by the fluctuations of microhabitats. Nevertheless, habitat preferences and requirements may change across seasons, as in the case of juveniles that select microhabitats with slightly different conditions in different times (Figs. 2E and 2F).

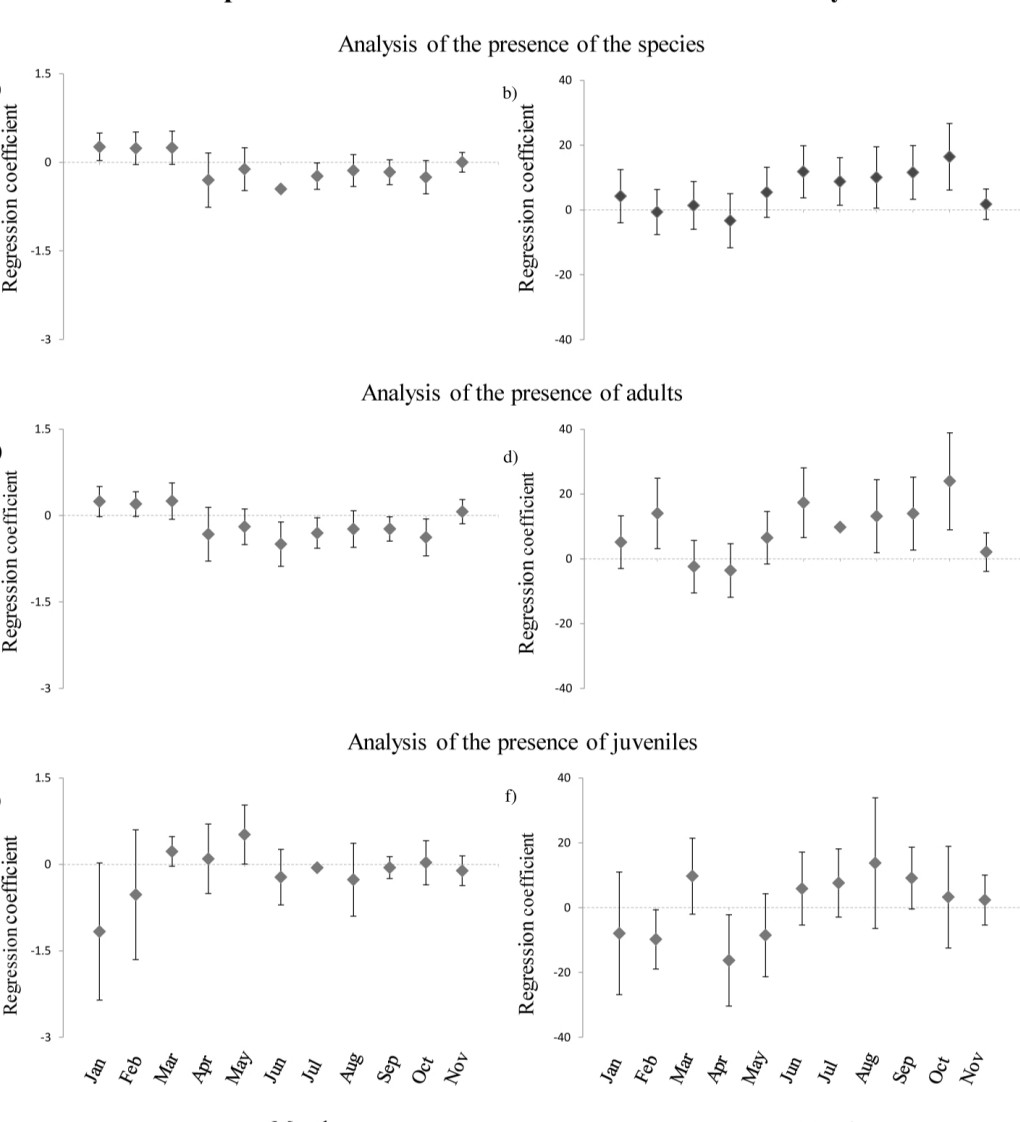

**Figure 2 Annual variation of the coefficients of regressions between presence/absence of cave salamanders, temperature and humidity.** (A)–(B): results of regression models analyzing all individuals encountered; (C)–(D) results of models analyzing adults only (E)–(F) results of models analyzing juveniles only. Results for December were not reported due to small sample size.

Cave depth represented the major gradient along with microhabitat features varied: as expected, humidity always increased and light decreased in the deepest sectors. The relationship between temperature and depth was more complex. During winter a positive relationship between temperature and depth was observed, while the relationship became negative during the warm months (Fig. 1A). Furthermore, all cave abiotic features (temperature, humidity and light) followed the variation of external conditions, which indeed were the major cause of fluctuations of internal microhabitats. While this influence

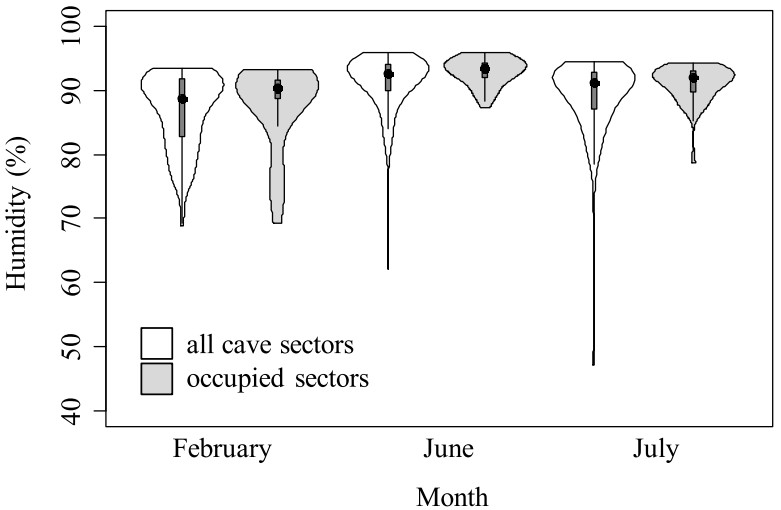

**Figure 3 Violin plots representing humidity in cave sectors available (white) and occupied by cave salamanders (grey), during three months.** The area of violin plots represents the distribution of cave sectors according to microclimate feature. Width of plots is proportional to the number of sectors showing such microclimate condition. The black points represent the medians, the grey boxes represent the second and third quartiles. The violin plots for temperature are available in Fig. S2.

**Table 5 Equivalency of species-habitat relationships (measured as Shoener's D) observed in different months.** Pairs of months for which the species-habitat relationships were not equivalent (after Bonferroni's correction: $\alpha' = 0.0083$) are in bold.

| | Temperature | | | Humidity | | |
|---|---|---|---|---|---|---|
| (A) All individuals | | | | | | |
| | Feb | Jun | Jul | Feb | Jun | Jul |
| Jan | 0.917 | 0.684 | 0.771 | 0.520 | 0.695 | 0.762 |
| Feb | | 0.616 | 0.639 | | **0.336** | **0.375** |
| Jun | | | 0.832 | | | 0.770 |
| (B) Adults only | | | | | | |
| | Feb | Jun | Jul | Feb | Jun | Jul |
| Jan | 0.844 | 0.644 | 0.703 | 0.650 | 0.595 | 0.612 |
| Feb | | 0.704 | 0.795 | | 0.650 | 0.601 |
| Jun | | | 0.790 | | | 0.650 |
| (C) Juveniles only | | | | | | |
| | Feb | Jun | Jul | Feb | Jun | Jul |
| Jan | 0.807 | 0.688 | 0.601 | 0.234 | 0.706 | 0.693 |
| Feb | | 0.528 | 0.428 | | **0.077** | **0.069** |
| Jun | | | 0.700 | | | 0.950 |

was strongest in the first meters of the caves, it remained clearly detectable at depths >20 m (Fig. 1), and therefore influenced the conditions experienced by salamanders.

During our surveys, detection probability of *Hydromantes italicus* was high, allowing us to obtain many observations, which are a necessary prerequisite for any habitat association

study. The observation of *H. italicus* was strongly related to time of survey. As observed in other studies (*Ficetola, Pennati & Manenti, 2012*; *Lunghi, Manenti & Ficetola, 2014*), salamanders were strongly associated with sectors characterized by specific microhabitat features, such as high humidity, low light and abundant spiders. Individuals showed differences in their response to abiotic features, which resulted in a different distribution of salamanders inside caves (*Ficetola, Pennati & Manenti, 2013*). Adults were associated with the wettest microclimates, while juveniles were present in apparently more stressful sectors as they were also present in sectors with lower humidity and less suitable temperatures. Such presence of juveniles also in suboptimal microhabitats has been observed also in other species of cave salamanders (*Ficetola, Pennati & Manenti, 2013*), and may allow juveniles to exploit more superficial environments, where they can find more food.

Beside some differences in habitat selection between adults and juveniles, a strong interaction between temperature, humidity and time of survey was consistently observed in most analyses (Tables 3, 4 and Fig. 2). For instance, salamanders tended to be associated to the coldest and wettest sectors of caves, but this pattern was not evident during late winter/spring (Figs. 2A and 2B). Such heterogeneity in habitat selection may occur both because individuals select different conditions during different times or life stage (selection change hypothesis) or because of the strong variability of available microhabitat conditions (environmental change hypothesis). In principle, it might be also possible that in certain periods juveniles are forced to move toward suboptimal areas because of competition with adults. However, this explanation is unlikely: previous studies explicitly testing this hypotheses have found evidence that juveniles are not displaced by adults (*Ficetola, Pennati & Manenti, 2013*), while behavioral analyses suggested lack of competition for territories (*Berti & Corti, 2010*).

Our data mostly support the environmental change hypothesis. First, the temperature gradient showed a clear inversion through the seasons (Fig. 1A). If salamanders always select the same optimal temperature (about 10–15 °C; Fig. S2), they can only find such conditions in the deepest sectors of caves, in which temperature is relatively warm during winter, and coolest during summer. Actually, most of equivalency tests were not significantly different from random expectations, indicating that the species consistently selected the same microhabitat. In other words, apparent changes in species-habitat relationships (e.g., positive relationship with temperature in winter and negative relationship in summer) occurred because the habitat occupied by salamander remained the same, but environmental gradients changed through the time. As a consequence, the relationships between microclimatic conditions and salamanders were not constant with time: in summer individuals tended to select the coldest, most humid sectors of caves, while the relationship was different during winter months (Fig. 2 and Table 5). In practice, selection of the same habitat resulted in regression coefficients that were remarkably different among seasons (Fig. 2). The difficulty of extrapolating regression results and linear relationships beyond the limits of environmental gradients tested is a major issue in ecological modelling (*Randin et al., 2006*; *Zurell, Elith & Schroder, 2012*). In principle, only sampling the whole spectrum of potential habitat conditions may allow

a full reconstruction of habitat preferences, but this is not feasible in the real world, because the available environmental gradients generally cover a limited range of conditions (*Soberon & Nakamura, 2009*; *Elith, Kearney & Phillips, 2010*).

Most of variation in species-habitat relationships was likely caused by the seasonal variation of temperature and humidity. Nevertheless, particularly in the analysis of humidity with juveniles, tests of niche equivalency between late winter and summer months were consistently rejected (Table 5). Cave salamanders are able to exploit the whole cave; therefore, if salamanders just require optimal abiotic conditions they can remain in farthest sectors where suitable microclimate is more stable. Conversely, in this study, salamanders during summer were associated to more humid sectors than in winter. This suggests a higher tolerance for dry sectors during winter, and supports the selection change hypothesis. Multiple, non-exclusive explanations are possible for such selection change. First, newborns *Hydromantes* normally hatch at the end of summer (*Lunghi et al., 2014*). Therefore, in the following winter, acquiring energy is a major priority for juveniles. The most superficial cave sectors are the ones with driest microclimate (Table 2), but show the highest abundance of prey. Actually, in our study caves, the potential prey richness (calculated as the summed N of species of Araneae (excluding *M. menardi*) and Diptera, as these taxa are the major food items for cave salamanders (*Vignoli, Caldera & Bologna, 2006*; *Crovetto, Romano & Salvidio, 2012*)) quickly decreases with depth (generalized linear model with Poisson error, taking into account month of survey: $B \pm SE = -0.024 \pm 0.006$, $\chi^2_1 = 201.3$, $P < 0.0001$). This indicates that juveniles may trade-off microclimatic optima for food availability (*Vlachos et al., 2014*). Actually, the end of winter may be a particularly important period, as in this period many invertebrates end their winter latency (*Bale & Hayward, 2010*). Efficient exploitation of seasonal peaks of food resources may be a key of fast development during the first years. Furthermore, the negative consequences of low humidity may be stronger in summer. Low environmental temperature reduces metabolism in ectotherms, which limits oxygen needs. As lungless salamanders exchange gasses mainly through their skin, and the efficiency of this skin function increases with high level of moisture (*Spotila, 1972*), during the cold season the individuals could be more tolerant to low humidity because of their lower respiration needs.

The peculiar physiology of plethodontids forces these salamanders to live within very narrow typologies of habitat. However, under certain circumstances, individuals may select conditions that are closer to their physiological limits (*Kearney et al., 2013*). This is likely the case for juveniles. Underground environments suffer constant food scarcity (*Romero, 2009*), but juveniles require consistent food supply in order to grow and reach maturity. Scarce access to food resources during juvenile stages poses major constraints on development, and may have prolonged consequences and even impact lifetime fitness (*Wong & Kölliker, 2014*). Therefore, in certain months, young salamanders exploit superficial sectors with more stressful abiotic conditions, but they receive enough food input from the outdoor environment to offset the risk.

In principle, the "optimal" habitat of a species should match species requirements for multiple parameters, ranging from metabolism to water balance and food availability.

However, such "ideal" conditions are rarely available in the real world, and species have to deal with environmental variability, which causes frequent changes of habitat conditions and resources availability (*Seebacher & Alford, 1999*; *Araújo et al., 2010*; *Fredericksen, 2014*). Our study explores the complexity of habitat use patterns under variable conditions, and highlights difficulties in determining habitat selection processes. When necessary resources are inversely correlated along environmental gradients, habitat choice will be the results of a trade-off between the multiple requirements of a species. We showed that such trade-off may be not constant with time or life stage, as both species priorities and habitat features may change across time. Individuals often require different resources depending on their life stage, and thus must shift their habitat selection to exploit different environments to satisfy their needs (*Cox & Cresswell, 2014*; *Dittmar et al., 2014*; *Webb et al., 2014*). Habitat selection studies are often based on data collected over temporal snapshots. However, seasonality is a pervasive feature of natural environments, highlighting the importance to always take into account the potential seasonal variation and considering the interactions between the requirement of individuals and the variability of habitats.

## ACKNOWLEDGEMENTS

We thank two reviewers for constructive comments on a previous version of this manuscript. GFF belongs to the Laboratoire d'Ecologie Alpine, which is part of Labex OSUG@2020.

### Funding

GFF was funded by Labex OSUG@2020 (Investissements d'avenir—ANR10 LABX56). The funders had no role in study design, data collection and analysis, decision to publish, or preparation of the manuscript.

### Grant Disclosures

The following grant information was disclosed by the authors:
Labex OSUG@2020 (Investissements d'avenir—ANR10 LABX56).

### Competing Interests

The authors declare there are no competing interests. Enrico Lunghi is representing the Association Natural Oasis as President.

### Author Contributions

- Enrico Lunghi conceived and designed the experiments, performed the experiments, analyzed the data, contributed reagents/materials/analysis tools, wrote the paper, prepared figures and/or tables, reviewed drafts of the paper.
- Raoul Manenti conceived and designed the experiments, contributed reagents/materials/analysis tools, wrote the paper, reviewed drafts of the paper.
- Gentile Francesco Ficetola conceived and designed the experiments, analyzed the data, contributed reagents/materials/analysis tools, wrote the paper, prepared figures and/or tables, reviewed drafts of the paper.

## Animal Ethics

The following information was supplied relating to ethical approvals (i.e., approving body and any reference numbers):

University of Florence approved the project by regular department's application and we followed our institutional guidelines.

## Field Study Permissions

The following information was supplied relating to field study approvals (i.e., approving body and any reference numbers):

Apuan Alps Regional Park (no 5, 4/04/2013),

District of Prato (no 448, 2013),

District of Pistoia (no 0022597/2013/P),

District of Lucca (no 731, 21/02/2013).

## Data Availability

The following information was supplied regarding the deposition of related data:

Raw data can be found in the Supplemental Information.

## Supplemental Information

Supplemental information for this article can be found online at http://dx.doi.org/10.7717/peerj.1122#supplemental-information.

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
