# Peer review of "Seasonal variation in microhabitat of salamanders: environmental variation or shift of habitat selection?"

_PeerJ, doi:10.7717/peerj.1122_

## Round 0.1 · original submission · Major Revisions

Please revise your paper according to the comments of the reviewers. Note that it may have to be re-reviewed and may not necessarily be accepted for publication even after revision.

Reviewer 1 ·

Basic reporting

The article is written in English although I suspect that English is not the authors primary language. Many terms used are ambiguous with regards to meaning and need to be explained. For example, Depth is a variable that is examined in this paper however the authors seem to mean distance from cave aperture. Similarly, habitat and environment both appear in the discussion and I am uncertain if the authors are referring to the same thing or not and indeed if they are using both interchangeably with the microhabitat variables that they examined in the paper. I would also suggest that the authors reexamine their paper to ensure that they are truly meaning preference/choice vs use/usage. I have made numerous comments to the word document with suggested terminology changes or to seek clarification. I would suggest that the authors work with a native English speaker to re-write and add clarity to their manuscript prior to re-submission.
The figures are appropriate and I especially like the violin graphs. However, some of the figures need additional attention. Figure legends should be clarified on some examples; several x-axis refer to Time(Month) where I believe that Month is the more accurate label; and graphs often use abbreviations in their legends and labels where there would be plenty of space to spell out the variable. Additionally, the font in the figures is different than that of the manuscript (I am unsure on PeerJ policies regarding this matter but I believe it would improve the looks of the paper to maintain the same font,.
I reviewed the literature cited and there are no missing citations, nor are any papers listed that are not cited in text. The citations are relevant and refer to recent findings in the field. Word choice is often imprecise and ambiguous.

Experimental design

The research questions are original to the authors and the research questions are clearly defined. Unfortunately, I cannot discern the rigor of the study as much information appears to be vague or missing. Examples of this include sample sizes. The paper mentions 15 caves sampled 12x each over one year for a total of 180 samples of 1087 salamanders. However, much of the analysis is further subdivided into separate seasonal aspects and of different age classes of individuals. These numbers are not provided. For one analysis the number of juveniles is provided but not of adults and thus total numbers of individuals. Similarly, the authors broke the caves up into 3 m sections for sampling stating that they sampled as deep (really distance from entrance) as they could without needing speleological equipment. Figure 1 presents variables at a depth of up to 21 m. However, it is not explained if this is the deepest sector that all caves were sampled to (with some caves going deeper), or if this was the deepest depth at which at least one cave was sampled. Thus the bar denoting 3 m surely represents 15 caves but the 21m bar may represent anywhere from 1 to 15 caves.
Further, I question why depth was not considered as a factor influencing either detection or occupancy.

Validity of the findings

It is difficult to determine the validity of the findings because much information such as sample sizes are missing. Further, I question whether the authors have truly examined one of the hypotheses they purport to test.
The authors seek to examine whether Italian Cave Salamander micro-habitat occupancy better follows the selection change hypothesis (where salamanders change their habitat usage seasonally or by age/stage class) or the environmental change hypothesis (where distribution is altered by temporal environmental changes). They suggest that their results mainly follow the environmental change hypothesis. Mainly is used because, juveniles were found in less humid environments during the winter months suggesting the selection change hypothesis. They suggest that this may be related to arthropod abundance, but do not present this data as it will be the basis of a separate paper leaving the reviewer to speculate as to its validity. If this is the case, I would suggest that the paper on Meta spider micro-habitat selection be published first and referenced in this study. Another potential reason for this would be juvenile being more tolerance of less humid conditions in the winter given lower metabolic needs. I would posit that in the absence of any spatial or behavioral data, it is just as possible that juveniles were excluded from “preferred” environments by adults.
Of interest and possibly confounding the results in examining the environmental change hypothesis is the fact that temperature inversion that occurs seasonally. The warmest areas of the cave in the winter become the cooler areas of the caves in the summer and these same areas are always have the lowest light readings and highest Relative Humidity (Figure 1). This temperature inversion results in the deepest sectors of the cave being the most stable environments in the cave and the authors indicate that it also exhibits the preferred temperature profiles for salamanders. This results in more stable conditions seasonally in one area of the caves. It is possible therefore that there is no change in salamander distribution because, while the environmental conditions are changing throughout the cave, there is no change in distribution.
Ideally, salamanders would have been marked and the various sectors they inhabited would have been recorded in each subsequent monthly survey. However, the authors could still examine the effect of sector depth on salamander presence. This variable should also be included in determining the detection estimate as deeper and darker, (possibly narrower, wetter, deep drop-offs and other things that would necessitate the use of speleological equipment) could possibly effect the researchers' ability to detect the species.

Annotated reviews are not available for download in order to protect the identity of reviewers who chose to remain anonymous.

Reviewer 2 ·

Basic reporting

This article needs to be edited for grammar. It is readable, but there are multiple areas with issues that lead to ambiguities.

Authors provide a clear explanation of the hypotheses they are testing. However, consider restructuring the introduction to draw more attention to the hypotheses names (environmental change and selection change). Pulling them out of the parenthesis would make it more obvious they are central concepts that will be returned to. Additionally, a literature review specific for plethodontids and the questions focused on in this study would be helpful. From the introduction it appeared very little to nothing had been done, but from the methods it became more apparent that the environmental selection hypothesis had been tested in cave salamanders.

Figures. What do each of the elements in the violin plot represent (bars, violin length, width, etc)?

Experimental design

The design generally appears robust, but additional details are required to fully assess the methods, and in particular the analysis. I am confident the authors can address these issues. Please see my detailed comments for specific issues.

Validity of the findings

Without clarification of some of the analysis and additional results, I cannot assess the validity of the findings. The authors refer to AICc analyses multiple times, but the only values presented are for one analysis and they are relegated to the supplementary material. I would recommend 1) moving the AICc analyses table to the main text. 2) I would like to see tables similar to S1 for all other hypotheses tested with AICc so readers can assess the authors’ findings for themselves. 3) What was the rationale for not testing single factor models, a null model or a global model? For example, you could directly test your assertion on lines 329-330 that most variation is likely caused by two single abiotic variables. Unless there is a strong reason from not including these models in the set, I would like to see them included.

Additional comments

Please see attached comments for line by line review

Annotated reviews are not available for download in order to protect the identity of reviewers who chose to remain anonymous.

---

## Round 0.2 · accepted · Accept

Thank you for your careful attention to detail.